# Comparative Proton and Photon Irradiation Combined with Pharmacological Inhibitors in 3D Pancreatic Cancer Cultures

**DOI:** 10.3390/cancers12113216

**Published:** 2020-10-31

**Authors:** Josephine Görte, Elke Beyreuther, Erik H. J. Danen, Nils Cordes

**Affiliations:** 1OncoRay—National Center for Radiation Research in Oncology, Faculty of Medicine Carl Gustav Carus Technische Universität Dresden, 01307 Dresden, Germany; Josephine.Goerte@uniklinikum-dresden.de (J.G.); Elke.Beyreuther@uniklinikum-dresden.de (E.B.); 2Institute of Radiooncology—OncoRay, Helmholtz-Zentrum Dresden—Rossendorf, 01328 Dresden, Germany; 3Institute of Radiation Physics, Helmholtz-Zentrum Dresden—Rossendorf, 01328 Dresden, Germany; 4Division of Drug Discovery and Safety, Leiden Academic Centre for Drug Research, Leiden University, 2333CC Leiden, The Netherlands; e.danen@lacdr.leidenuniv.nl; 5German Cancer Consortium, Partner Site Dresden: German Cancer Research Center, 69120 Heidelberg, Germany; 6Department of Radiotherapy and Radiation Oncology, University Hospital Carl Gustav Carus, Technische Universität Dresden, 01307 Dresden, Germany

**Keywords:** PDAC, radiotherapy, photon irradiation, proton beam irradiation, 3D cell culture, molecular targeting, DNA repair, NHEJ

## Abstract

**Simple Summary:**

Due to higher precision and consequent sparing of normal tissue, pancreatic cancer patients might profit from proton beam radiotherapy, a treatment modality increasingly used. Since molecular data upon proton irradiation in comparison to standard photon radiotherapy are limited in pancreatic cancer, the aims of our study were to unravel differences in the effectiveness of photon versus proton irradiation and to exploit radiation type-specific molecular changes for radiosensitizing 3D PDAC cell cultures. Although protons showed a slightly higher effectiveness and a stronger induction of molecular alterations than photons, our results revealed a radiation-type independent sensitization of molecular-targeted agents selected according to the discovered molecular, radiation-induced alterations.

**Abstract:**

Pancreatic ductal adenocarcinoma (PDAC) is a highly therapy-resistant tumor entity of unmet needs. Over the last decades, radiotherapy has been considered as an additional treatment modality to surgery and chemotherapy. Owing to radiosensitive abdominal organs, high-precision proton beam radiotherapy has been regarded as superior to photon radiotherapy. To further elucidate the potential of combination therapies, we employed a more physiological 3D, matrix-based cell culture model to assess tumoroid formation capacity after photon and proton irradiation. Additionally, we investigated proton- and photon-irradiation-induced phosphoproteomic changes for identifying clinically exploitable targets. Here, we show that proton irradiation elicits a higher efficacy to reduce 3D PDAC tumoroid formation and a greater extent of phosphoproteome alterations compared with photon irradiation. The targeting of proteins identified in the phosphoproteome that were uniquely altered by protons or photons failed to cause radiation-type-specific radiosensitization. Targeting DNA repair proteins associated with non-homologous endjoining, however, revealed a strong radiosensitizing potential independent of the radiation type. In conclusion, our findings suggest proton irradiation to be potentially more effective in PDAC than photons without additional efficacy when combined with DNA repair inhibitors.

## 1. Introduction

Pancreatic cancer, with pancreatic ductal adenocarcinoma (PDAC) being the most common subtype, presents one of the top tumor types with unmet needs and only minimal advances in treatment outcome in recent years [1,2]. PDAC is the fourth leading cancer in tumor-related deaths and despite extensive research advances, the 5-year survival is still poor, with 7% [2]. Tumor resection is the only curative regimen, however, only less than 20% of the patients profit from it. The challenging high therapy resistance of PDAC arises from various properties, such as a highly desmoplasmic microenvironment, genetic mutations and late diagnosis at already advanced or metastasized stages [2,3].

Due to the delicate localization of the pancreas close to the gastrointestinal tract, radiotherapy with photons is frequently used for metastatic PDAC or palliatively [2]. With great technical advances, proton beam therapy is, however, only applied at specific clinics worldwide. Based on its physical properties, treatment with proton beam radiotherapy confers better sparing of organs at risk and is currently discussed as a preferable treatment modality in comparison to standard photon radiotherapy [4]. It is the more optimal dose-depth profile of the proton beam that provides a higher precision. This is characterized by a low entrance dose and an increase at the stop, the so-called Bragg peak. The Bragg peak can be spread out (spread-out Bragg peak, (SOBP)) by combining various beams, each with a different initial energy, over the tumor volume and positioning them precisely [5,6].

Despite the first promising clinical results obtained for proton beam therapy of other tumor entities, little is known for PDAC. It further remains open whether proton irradiation induces similar or different modifications to the DNA and to signal transduction events and whether such differences might be exploitable for combinatorial therapies with biologicals [7,8]. In general, combining photon and proton irradiation with DNA repair inhibitors appears promising and may be superior to targeting of receptor tyrosine kinases (RTK) or cytoplasmic protein kinases, as the DNA remains as most important target in the context of cell survival after exposure with ionizing radiation.

To comparatively elucidate the efficacy of proton versus photon irradiation and to identify novel potential therapeutic targets for these two radiation types in PDAC, we cultured PDAC cell lines under more physiological three-dimensional (3D), matrix-based conditions and examined their tumoroid formation capacity and phosphoproteome modifications. Our results indicate a cell-line-dependent higher efficacy for protons over photons regarding 3D PDAC tumoroid growth. Combination treatments with RTK, cytoplasmic protein kinase and DNA repair inhibitors showed an inhibitor- and cell-line-specific cytotoxic and radiosensitizing potential, which presented as equipotent for photons and protons and with a strong non-homologous endjoining (NHEJ) dependency in our PDAC cell line panel.

## 2. Results

### 2.1. Sensitivity towards Photon and Proton Irradiation Varies in Human 3D PDAC Cell Cultures

We commenced our study by a comparative tumoroid growth analysis upon photon and proton irradiation in a panel of five human PDAC cell lines grown in physiological laminin-rich extracellular matrix (lrECM) (Figure 1A). While basic tumoroid formation was similar for the experimental photon and proton irradiation setup, we found varying intrinsic radiosensitivities towards photon irradiation, with MiaPaCa-2 being the most sensitive and Colo357 the most resistant cell line (Figure 1B–D; values for relative biological effectiveness (RBE) and surviving fraction of 2 Gy (SF2) are shown in Appendix A, respectively). In general, the degree of radiosensitivity towards photon irradiation differs to the one found for proton irradiation. Apart from BxPC-3 cells, tested PDAC cell lines demonstrated either a significantly higher sensitivity to protons than photons or showed a trend towards higher sensitivity to protons that resulted in reduced tumoroid growth (Figure 1D). Collectively, our data revealed a greater reduction in PDAC tumoroid growth upon proton than photon irradiation.

### 2.2. Proton Irradiation Elicits Greater Changes in the Phosphoproteome than Photon Irradiation

We next sought to unravel potential differences in the molecular mechanisms causing the observed differential effectiveness between proton and photon irradiation in PDAC cell cultures. Hence, we performed a screening in form of a broad-spectrum phosphoproteome analysis (Figure 2A). 3D lrECM grown Colo357 and MiaPaCa-2 cell lines were chosen as models due to differences in their intrinsic radiosensitivity towards photon irradiation (MiaPaCa-2 most sensitive, Colo357 most resistant). Generally, the phosphorylation patterns differed between proton and photon irradiation (Figure 2B,C, Appendix A). Interestingly, we detected more increases and decreases in phosphorylation in both 3D PDAC cell cultures upon proton irradiation than photons according to our cut-offs of ≥50% and ≥30%, respectively (Figure 2D). Among the detected 584 phosphorylation sites of 331 proteins from the array, 6 out of 37 and 10 out of 34 showed an overlapping ≥30% decrease between protons and photons in Colo357 and MiaPaCa2 cells (Figure 2E). A common ≥50% increase after proton and photon irradiation was detectable for 10 out of 69 and 23 out of 76 protein phosphorylation sites in the two tested cell lines (Figure 2E).

A more rigorous analysis focused on phosphorylation sites with an increase of 50% or higher, aiming to identify potential radiosensitizing targets specific to protons, photons or both. For photons, Merlin (p-Ser10) was the only shared unique candidate among 39 phospho-sites in 3D Colo357 and MiaPaCa-2 cell cultures (Appendix A). Upon proton irradiation, we found Catenin β (p-Ser33), Smad2/3 (p-Thr8) and GSK3 α (p-Ser21) among 68 unique phospho-sites in both cell lines (Appendix A). Overlapping among 30 phospho-sites for proton and photon irradiation were p53 (p-Ser6), Synuclein (p-Tyr125) and VASP (p-Ser157) (Appendix A). Given the fact that these protein phosphorylations either inhibit the enzymatic activity (GSK3α), induce ubiquitination (Merlin) or lack specific, clinically relevant inhibitors (Synuclein, VASP), we decided to analyze our data on the protein level instead of the phosphorylation level (≥50% increase; Figure 2F).

In summary, proton irradiation led to greater changes in phosphorylation events than photon irradiation in 3D PDAC cell cultures, and our analysis revealed a strikingly different pattern of phosphorylation events for proton and photon irradiation.

### 2.3. Unique Phosphoproteomic Changes after Photon and Proton Irradiation Fail to Be Exploitable for Radiosensitization

The list of all proteins showing a ≥50% increase in phosphorylation that entailed enhanced enzymatic activity prompted us to search for both radiation-type-specific and non-specific targets (Figure 2F). Out of this set of proteins, we chose druggable targets for which inhibitors are already clinically applied or are being tested in clinical trials (Figure 3A). We selected one target that was specifically altered after either photon irradiation (estrogen receptor α, ER-α), proton irradiation (HER2), or both photon and proton irradiation (Chk1) (Figure 3A). Hydroxy-tamoxifen was used to deactivate ER-α, the inhibitory antibody trastuzumab (Ontruzant^®^, MSD SHARP & DOHME GMBH, Haar, Germany) and the small molecule inhibitor lapatinib were applied for HER2 and prexasertib, a small molecule inhibitor, for Chk1.

Concerning basal tumoroid growth after treatment, we found only moderate effects in both analyzed cell lines, Colo357 and MiaPaCa-2 (Figure 3B). In combination with irradiation, however, Tamoxifen enhanced Colo357 and MiaPaCa-2 cell sensitivity towards 6-Gy photons by 1.2- and 1.6-fold, respectively (Figure 3C). The sensitivity towards 6-Gy protons was only elevated in MiaPaCa-2 cells by 1.4-fold (Figure 3C). In contrast, trastuzumab failed to radiosensitize the tested PDAC cell lines to photons and protons (Figure 3C). In contrast, treatment with lapatinib significantly sensitized both analyzed cell lines towards irradiation with photons and protons (Appendix A). The enhancement ratios were greater for Colo357 cells (1.3 for photons; 1.4 for protons) than for MiaPaCa-2 (1.1 for photons; 1.2 for protons) (Appendix A). Intriguingly, prexasertib elicited radiosensitization to photons and protons both in Colo357 and MiaPaCa-2 cells. The resulting enhancement ratios for Colo357 and MiaPaCa-2 cells were 1.5 and 1.6, respectively, for photons and 1.3 in both cell lines for protons (Figure 3C and Appendix A). Our findings suggest targeting of ER-α and Chk1 to convey slightly higher radiosensitizing efficacy to photons in relation to protons (Figure 3D), whereas targeting of HER2 tends to increase the efficacy for protons (Figure 3D and Appendix A). The phosphorylation and expression of these targeted proteins after irradiation remained unchanged for ER-α and HER2 (Appendix A) in contrast to Chk1, showing a hyperphosphorylation at its activating phospho-site S345 1 h after irradiation with both photons and protons (Appendix A).

Taken together, despite detectable changes in protein phosphorylation in the phosphoproteome analysis, we were unable to therapeutically exploit these differences uniquely for PDAC cell sensitization to either photon or proton irradiation.

### 2.4. Screening of Signal Transduction and DNA Repair Inhibitors Reveals Similarity in the Radiosensitizing Potential for Photon and Proton Irradiation

We subsequently conducted a screen with inhibitors deactivating either RTK signal transduction (EGFRi, MAPKi, MEKi, PI3Ki, VEGFRi) or DNA repair processes (ATMi, DNA-PKi, MDM2i, Mre11/Rad50/Nbs1 protein complex (MRNcomplexi), PARPi, Rad51i). While 3D tumoroid growth of Colo357 cells was diminished upon EGFRi, VEGFRi and Rad51i, MiaPaCa-2 tumoroid growth remained unaffected (Figure 4A). Cell-line-dependently, signal transduction inhibitors elicited 3D tumoroid growth reduction in both tested cell lines upon irradiation (Figure 4B and Appendix A). PI3Ki showed the strongest effect in both cell lines (Figure 4B and Appendix A). Interestingly, Colo357 and MiaPaCa-2 cells responded differently to MRNcomplexi upon proton irradiation relative to photons (Figure 4B and Appendix A). Pretreatment with Rad51i led to a radioprotection in MiaPaCa-2 cells (Figure 4B and Appendix A). In contrast, PARPi, DNA-PKcsi and ATMi were the most efficient regarding reduction in tumoroid growth (Figure 4B and Appendix A). The similarity of the radiosensitizing efficacy of the different signal transduction and DNA repair inhibitors for photons and protons is visualized in Figure 4C.

In conclusion, we found effective signal transduction inhibitors such as PI3Ki and DNA repair inhibitors as radiation type-independent radiosensitizers. Notably, the screen pointed at specific DNA repair inhibitors associated with NHEJ.

### 2.5. NHEJ Dependency Is Exploitable for PDAC Cell Sensitizitation to Protons and Photons

The DNA damage response, in general, plays a pivotal role in tumor cell survival and the associated DNA repair enzymes are considered as ideal therapeutic targets. We next sought to confirm the different DNA repair inhibitor efficacies for radiosensitization in our larger PDAC cell line panel. Single pharmacological inhibition of Rad51, PARP, DNA-PK and ATM indeed echoed the results observed in the Colo357/MiaPaCa-2 cell model screen (Figure 5A). Overall, and similar to our screen, we discovered the following increasing 3D PDAC tumoroid reduction for photons as well as protons: Rad51i < PARPi < DNA-PKcsi < ATMi (Figure 5B and Appendix A). Although differing in their absolute values among the tested PDAC cell lines, the discrepancies between photons and protons turned out to be minor, ranging between approximately −0.2 and 0.3 (Figure 5C). This notion is also visualized in Figure 5D, in which the enhancement ratios of all conditions are plotted. Moreover, these results are underpinned by taking together the differences in the radiation response to photons and protons of the whole PDAC cell line panel. Indicated by Δ values close to zero, the reduction in tumoroid growth after HR and NHEJ inhibition presented as largely independent from radiation type (Figure 5E).

Conclusively, the data generated in our PDAC cell line panel revealed the classical and alternative NHEJ DNA repair pathways as candidate targets for sensitization to proton- and photon-based radiotherapy.

## 3. Discussion

Given the higher precision accompanied by optimized sparing of normal tissue, proton beam irradiation is considered more favorable for PDAC patients than photon irradiation. However, supportive large clinical datasets, as well as systematic preclinical insights, are lacking. To address this point, we conducted a preclinical study in a panel of PDAC cell lines grown in 3D extracellular matrix with and without molecular-targeted pretreatments. Our study revealed an equal or higher efficacy of low-LET protons over photons in terms of reducing PDAC tumoroid formation. Moreover, we showed a greater extent of phosphoproteome alterations upon proton irradiation compared with photon irradiation. The targeting of proteins identified in the phosphoproteome uniquely altered by protons or photons failed to mediate marked radiation type-specific radiosensitization. Instead, inhibition of DNA repair proteins acting in NHEJ revealed a strong radiosensitizing potential independent of the radiation type.

These observations are in line with comparative survival analysis of proton versus photon irradiation in other tumor entities such as glioma stem cells [9,10] or lung cancer cells [11]. Slightly different results, however, were found in cells cultured under 2D conditions [12]. The higher efficacy of protons over photons reported in such studies might have been caused by differences in cell and nuclear size and chromatin organization; both parameters have been reported by us earlier for photon irradiation [13]. Intensive discussion about the RBE as a clinically relevant parameter requires in vivo growth conditions to be determined. Generally, the RBE for protons is considered to be 1.1, but recent studies have already demonstrated its variability [11,12,14,15], which is also reflected in the PDAC cell line panel presented here.

Taking into consideration the therapeutic exploitability of the hallmarks of cancer represented by altered prosurvival signal transduction and the associated opportunities for treatment personalization [16,17], we undertook a comparative broad-spectrum phosphoproteome analysis of the response to proton versus photon irradiation 1 h post-irradiation. To identify common targets in PDAC cells, the most sensitive and most resistant cell lines to photon irradiation were chosen, revealing only a minor overlap of protein phospho-site modifications. The obtained results largely defeated our aim to find uniquely altered proteins to exploit them as sensitizers in PDAC cell lines towards either photon or proton irradiation. Several reasons might be causative, like the chosen early snapshot at 1 h after irradiation, the limited number of proteins on the array or the limited number of cell models tested. Future, more systematic examinations are warranted.

Nevertheless, we were able to discover three druggable targets from the array. Targeting the ER-α, unique for photon irradiation, and Chk1, for both photon and proton irradiation, resulted in moderate but significant radiosensitization irrespective of the radiation type. The ineffectiveness of trastuzumab-mediated HER2 deactivation in PDAC cell models observed in our study is in line with both preclinical and clinical findings but has not been addressed in combination with radiotherapy to date [18,19]. As trastuzumab cytotoxicity has been connected to immune cells to induce antibody-dependent, cell-mediated cytotoxicity in PDAC cells [20], a second, non-antibody-based small molecule inhibitor, lapatinib, was applied prior to photon or proton irradiation. We hypothesize the higher efficacy of lapatinib, as compared with trastuzumab, to originate from its inhibitory spectrum against HER2 and EGFR. Concerning tamoxifen, our study is the first to identify a radiosensitizing potential towards both photon and proton irradiation in PDAC cells. For other cancer cell models, the role of ER-α is already known in the radiation response upon photon irradiation and attributed to its interactions with DNA repair proteins [21]. With regard to Chk1 inhibition, Vance et al. showed a sensitization of pancreatic cancer cell lines towards photon irradiation [22]. Likewise, Chk1 deactivation elicited radiochemosensitizing effects for photons together with gemcitabine [23]. The body of literature for combined treatment with protons and biologicals is limited. Only one report outlined a higher degree of sensitization towards protons than to photons after Chk1 inhibition in triple-negative breast cancer MDA-MB-231 cells [24]. Conversely, our results, especially the ones obtained in MiaPaCa-2 cells, indicate a higher efficacy of photons over protons.

As alternative approach, we widened the inhibitor spectrum, including candidates that are either validated molecular targets for other cancer types or are considered as potential cancer targets [25]. Beyond the targets identified from our phosphoproteome array, a screening with a panel of signal transduction and DNA repair inhibitors provided clear evidence for a superiority of DNA repair inhibitors over inhibitors for receptor tyrosine kinase or cytoplasmic protein kinases. In contrast to others demonstrating afatinib, a pan-ErbB inhibitor, to radiosensitize PDAC cell lines [26], we found no effect after specific ErbB2/HER2 targeting. This, however, is in line with the findings of Kimple et al. [27]. PI3K inhibition, still under debate as potential cancer target [28], did sensitize PDAC cells to photon and proton irradiation [29]. Although the phosphoproteome array encompassed a part of these proteins such as DNA-PK, EGFR, MEK1, MDM2, p38 MAPK and VEGFR1/2, these proteins neither fulfilled our definition in terms of a ≥30% decrease or ≥50% increase in phosphorylation, nor did they overlap in the two tested cell lines. Yet, inhibition of these proteins, except for MDM2 and p38 MAPK, resulted in a cell-line-dependent and similar sensitization to photon and proton irradiation. These findings propose that the function of a protein in the radiation survival response upon photons and protons is not necessarily reflected by the detectable radiation-induced changes in phosphorylation or activity. Future studies are warranted, which analyze time points after irradiation beyond our 1 h post-irradiation snapshot, presented here.

Intriguingly, our conducted experiments revealed the efficacy of the selected inhibitors to be independent from the radiation type in 3D lrECM PDACcultures. Our data pinpoint two aspects: (i) the radiation type-unrelated strong dependence of PDAC cell survival on DNA repair; (ii) the great similarity in dependence on the same DNA repair machinery upon photon and proton irradiation. DNA double-strand breaks are repaired by two main DNA repair pathways, i.e., homologous recombination (HR) and NHEJ [5,8]. Our 3D lrECM PDAC cell culture panel showed clear a dependence on classical and alternative NHEJ, as indicated by DNA-PKcs and PARP inhibitors, in contrast to HR impairment using a Rad51 inhibitor. Likewise, Li et al. reported PDAC cell lines to strongly depend on DNA repair via NHEJ when irradiated with photons [30]. In Panc-1 and KP4 cells, as well as in esophageal cancer cells, PARP inhibition resulted in sensitization towards proton irradiation [31,32]. In further studies in Ligase IV knockout mouse embryonic fibroblasts and DNA-PKcs-deficient glioblastoma cells, both resembling NHEJ deficiency, a similar radiosensitization to photons or protons was exhibited [12]. We and others already reported radiosensitization by ATM inhibition in photon-irradiated 2D-cultured PDAC cells [33,34]. Here, we document the sensitizing potential of ATM inhibition in a panel of 3D-cultured, proton-irradiated PDAC cell lines. Intriguingly, the inhibition of the MRN complex, consisting of MRE11, Rad50 and NBS1 [35], failed to mediate radiosensitization in contrast to ATM deactivation.

Concerning HR targeting, Colo357 cells were the only cell line strongly and marginally radiosensitized by Rad51 inhibition to photon and proton irradiation, respectively. Importantly, we were unable to confirm the reported greater cellular sensitivity towards protons under HR-deficiency in the here-presented 3D lrECM PDAC cell cultures [31,36,37].

## 4. Materials and Methods

### 4.1. Antibodies

All used primary antibodies for Chk1, phospho-Chk1 S345, Estrogen Receptor-α, phospho-Estrogen Receptor-α S118, HER2/ErbB2 and phospho-HER2/ErbB2 Y1248 were purchased from Cell Signaling (Frankfurt, Germany). Secondary antibodies anti-mouse IgG, HRP conjugated and anti-rabbit IgG, HRP conjugated, were purchased from Pierce (Bonn, Germany).

### 4.2. Cell Culture

Human pancreatic ductal adenocarcinoma (PDAC) cell lines BxPC3, MiaPaCa2, Panc-1, and Patu8902 were purchased from the American Type Culture Collection (ATCC). Colo357 cell lines were a kind gift from Chr. Pilarsky (University Erlangen-Nürnberg, Germany). The origin and stability of the cells were routinely monitored by short-tandem repeat analysis (microsatellites). All cell lines were cultured in Dulbecco’s modified Eagle medium (DMEM, Sigma-Aldrich, Taufkirchen, Germany) with 10% fetal calf serum (FCS, Sigma-Aldrich, Taufkirchen, Germany) and 1% non-essential amino acids (Sigma-Aldrich, Taufkirchen, Germany) at 37 °C with 8.5% CO_2_ at pH 7.4. In all experiments, asynchronously growing cells were used. All cells tested negative for Mycoplasma by using the mycoplasma detection kit Venor^®^GeM OneStep (Minerva Biolabs, Berlin, Germany).

### 4.3. 3D Tumoroid Formation Assay

PDAC cells were seeded for 3D tumoroid formation assay, as published [38]. In brief, cells were embedded into lrECM (Matrigel™; BD, Heidelberg, Germany) at a concentration of 0.5 mg/mL in 96 well plates coated with 1% agarose. Twenty-four hours later, cells were irradiated with 2, 4 or 6 Gy of photons or protons or left unirradiated. In case of inhibitor treatment, cells were pretreated for 1 h before irradiation. After 24 h, inhibitors (except for Trastuzumab) were removed and replaced by fresh complete DMEM. After a cell line-dependent incubation period of 7–13 days, PDAC tumoroids consisting of a minimum of 50 cells were counted microscopically.

### 4.4. Radiation Exposure

#### 4.4.1. Photon Irradiation

Cells were irradiated at room temperature using 2, 4, or 6 Gy single doses of 200-kVp X-rays (Yxlon Y.TU 320; Yxlon; dose rate ≈ 1.3 Gy/min at 20 mA) filtered with 0.5-mm Cu, as published [39]. The absorbed dose was measured using a Semiflex ionization chamber (PTW Freiburg; Freiburg, Germany). Cells in tissue culture plates were irradiated horizontally both for tumoroid formation assays (96-well plates) and whole-cell lysates (24-well plates) as published [39].

#### 4.4.2. Proton Irradiation

Proton irradiation (low-LET of 3.7 keV/µm) was performed at the horizontal fixed-beam beam line in the experimental hall of the University Proton Therapy Dresden (UPTD). For 150 MeV protons, a dedicated beam shaping system consisting of a double-scattering device and a ridge filter provides a laterally extended 10 × 10 cm^2^ proton field and a SOBP of 26.3 mm (90% dose plateau) in water. Cells in tissue culture plates were irradiated at room temperature using two different setting to assure mid-SOBP position. For tumoroid formation assays 96-well plates were placed perpendicular to beam axis (90°), whereas for whole-cell lysates 24-well plates were positioned at 42° relative to beam axis to avoid destruction of 3D structure [40]. For absolute dosimetry, a Markus ionization chamber (PTW) readout by an Unidos dosemeter (PTW) at sample position was applied. Details of absolute dosimetry and beam control are given in Beyreuther and colleagues [41].

#### 4.4.3. Calculation Relative Biological Effectiveness (RBE)

The RBE values were calculated as follows (1): 
(1)
$$RBE=\frac{D_{Photons}}{D_{Protons}}$$


The doses (D) were calculated as follows: data points were fitted in the linear-quadratic formula and the values for α and β were taken from the fitted curve. D was then solved at a SF of 50% from the following Formula (2): 
(2)
$$SF=\;e^{-\alpha D-\beta D^{2}}$$


### 4.5. Total Protein Extraction and Western Blotting

Whole-cell lysates, SDS-PAGE and Western blotting were performed as previously described in [39]. In brief, cells cultured in 0.5 mg/mL Matrigel were irradiated after 24 h or left unirradiated. One hour after irradiation, whole-cell lysates were harvested with RIPA lysis buffer containing a protease inhibitor (complete protease inhibitor cocktail from Roche, Mannheim, Germany) and phosphatase inhibitors (Na3VO4 and NaF from Sigma-Aldrich, Taufkirchen, Germany). After incubation for 30 min on ice, cells were mechanically lysed using a syringe followed by centrifugation at 13,000× *g* for 20 min to remove debris. The chemiluminescent detection was performed using ECL™ Prime Western Blotting System (Sigma-Aldrich). Densitometric analysis was carried out with ImageJ. In Appendix A, the original uncropped images of Western Blot are displayed.

### 4.6. Inhibitors and Reagents

Cells were treated with pharmacological inhibitors for ATM (KU55933, Calbiochem, San Diego, CA, USA; 10 µM) Chk1 (Prexasertib, Selleckchem, Houston, TX, USA; 1 and 3 nM), DNA-PK (NU7026, Selleckchem; 10 µM), EGFR (Tarceva^®^, Roche, Basel, Switzerland; 10 µM), ER-α (Hydroxy-Tamoxifen, Sigma Aldrich; 10 µM), HER2 (Ontruzant^®^, MSD SHARP & DOHME GMBH, Haar, Germany; 2 µg/mL and Lapatinib, Selleckchem; 1 µM), MAPK (SB203580, Selleckchem; 10 µM), MDM2 (AMG232, Axon Medchem, Groningen, The Netherlands; 10 µM), MEK (PD98059, Selleckchem; 20 µM), MRNcomplex (Mirin, Sigma Aldrich; 10 µM), PI3K (LY294002, Selleckchem; 10 µM), PARP (Olaparib, Cell Signaling, Frankfurt a. M., Germany; 10 µM), Rad51 (B02, Axon Medchem; 10 µM) and VEGFR (Axitinib, Sigma Aldrich; 1 µM) with the indicated concentrations. Ethanol (for Hydroxy-Tamoxifen), IgG (for Ontruzant) and DMSO (for all other inhibitors) were used as controls.

### 4.7. Phosphoproteome Analysis

Colo357 and MiaPaCa-2 cells were cultured in 0.5 mg/mL lrECM and exposed to 6-Gy photon or proton irradiation after 4 days or unirradiated. To determine differences in the early events on the molecular level, here the DNA damage response upon proton and photon irradiation, whole cell lysates were harvested 1 hour post-treatment as previously published [42]. The samples were transferred to Full Moon BioSystems Inc. on dry ice for conducting the Phospho Explorer Antibody Microarray. Proteins were biotin-labeled and put on preblocked microarray slides. Detection of total and phosphorylated proteins was carried out by the use of Cy3-conjugated streptavidin. The array consists of antibodies against 342 proteins and 606 phospho-sites. For analysis, protein phosphorylation was normalized to corresponding total protein expression. Proteins with phosphorylation site changes of at least a 30% decrease or 50% increase (arbitrary cut-off) were considered relevant and selected, and changes in both cell lines comparatively analyzed.

### 4.8. Statistics

Means ± standard deviation (SD) of at least three independent experiments were calculated. For statistical significance analysis of tumoroid formation capacity, two-sided Student’s *t*-test was performed with Excel (Microsoft) and a *p* value of less than 0.05 was considered statistically significant.

## 5. Conclusions

In summary, our comparative study unravels a cell-line-dependent higher efficacy of proton over photon irradiation. Phosphoproteome analysis revealed different phosphorylation patterns induced by photon and proton irradiation, the latter leading to more critical changes 1 h post-irradiation. Contrary to our hypothesis, we could not find any radiation-type-specific targets in this dataset that can be used for a clear radiosensitization in a cell line-dependent manner. Instead, we discovered that a radiosensitization mediated by the inhibition of specific kinases occurs to a similar extent for protons and for photons in a panel of PDAC cell line cultures under more physiological 3D, matrix-based conditions. Considering the translation of these results to the treatment of patients suffering from PDAC, further insights are required to better discriminate the differences in sensitivity towards proton versus photon irradiation. Despite the great benefit of higher precision, and sparing of normal tissue, our data fail to clearly document a benefit of the co-application of biologicals to proton beam irradiation over photon irradiation.

## Figures and Tables

**Figure 1 cancers-12-03216-f001:**
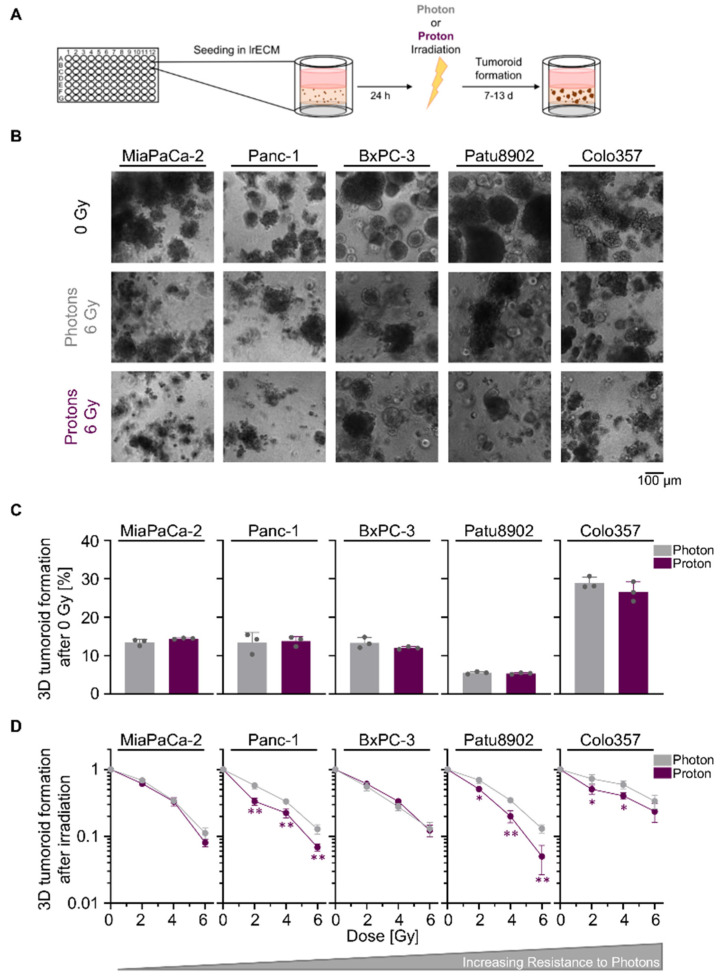
Proton irradiation tends to be more effective in reducing pancreatic ductal adenocarcinoma (PDAC) tumoroid growth than photon irradiation. (**A**) Experimental set-up for examining 3D PDAC tumoroid growth; (**B**) Representative bright-field images of unirradiated, 6-Gy photon and 6-Gy proton irradiated 3D PDAC tumoroids, scale bar: 100 µm; (**C**) 3D tumoroid formation capacity without irradiation; (**D**) 3D PDAC tumoroid growth after irradiation with 2, 4 or 6 Gy of photons and protons. Cell lines are ordered by increasing resistance to photon irradiation. Results show mean ± SD (*n* = 3; two-sided *t*-test; *, *p* < 0.05; **, *p* < 0.01).

**Figure 2 cancers-12-03216-f002:**
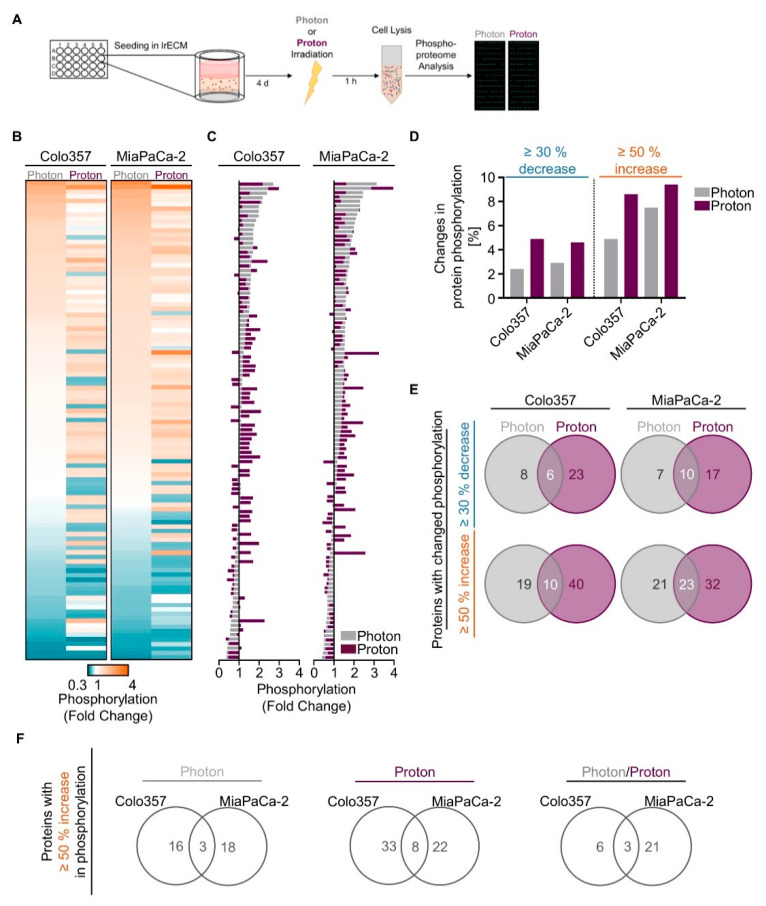
Changes in the phosphoproteome of 3D PDAC cell cultures by proton and photon irradiation. (**A**) Experimental set-up of phosphoproteome analysis; Alterations in phosphorylation sites upon irradiation are categorized into three ranges: (i) ≥30% decrease, (ii) ≥50% increase, (iii) ≤30% decrease + ≤50% increase. The latter is defined as ‘no change’ (see Appendix A). Data falling into the categories (i) and (ii) are plotted as (**B**) heatmaps and (**C**) waterfall blots after normalization to unirradiated controls (*n* = 1); (**D**) Percentage of phospho-sites among all detected 584 phospho-sites showing ≥30% decrease or ≥50% increase in phosphorylation after irradiation in indicated cell lines; (**E**) Venn diagram analysis indicating overlapping and specific alterations in phospho-sites after photon or proton irradiation; (**F**) Venn diagram analysis comparing proteins altered by ≥50% increase in phosphorylation of Colo357 and MiaPaCa-2 cell lines, either specifically after photons and protons or by both kinds of irradiation.

**Figure 3 cancers-12-03216-f003:**
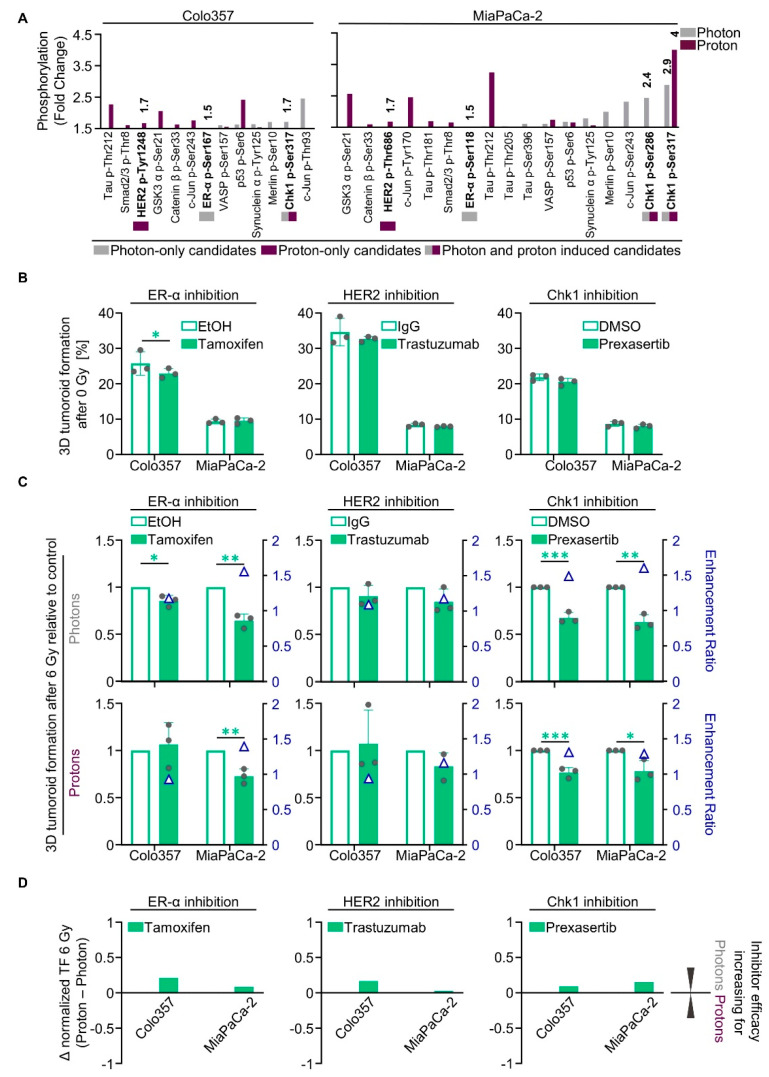
Photon- and proton-irradiated 3D PDAC cell cultures differentially respond to tamoxifen, trastuzumab and prexasertib. (**A**) Phospho-site alterations with ≥50% increase in phosphorylation (fold change; displayed data start at 1.5) in 6-Gy photon- or proton-irradiated Colo357 and MiaPaCa-2; (**B**) 3D tumoroid formation capacity of unirradiated Colo357 and MiaPaCa-2 cells treated with tamoxifen, trastuzumab or prexasertib (experimental set-up shown in Appendix A); (**C**) Normalized 3D tumoroid formation capacity upon 1-h pretreatment with tamoxifen, trastuzumab or prexasertib in combination with 6-Gy photon or proton irradiation. Enhancement ratio (tumoroid formation capacity after 6 Gy control treatment/tumoroid formation capacity after 6 Gy inhibitor treatment) are indicated as blue triangles; (**D**) Differences in the radiosensitizing efficacy of inhibitors visualized by Δ values of normalized tumor formation capacity (tumoroid formation capacity after 6 Gy of protons - tumoroid formation capacity after 6 Gy of photons). All results show mean ± SD (*n* = 3; two-sided *t*-test; *, *p* < 0.05; **, *p* < 0.01; ***, *p* < 0.001); Δ: delta, TF: tumoroid formation.

**Figure 4 cancers-12-03216-f004:**
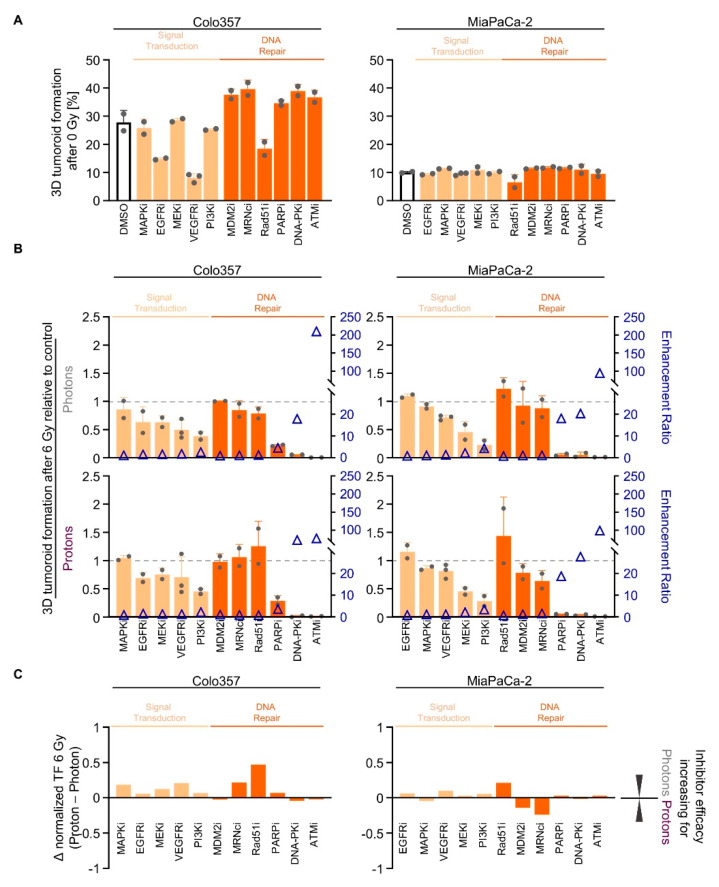
Targeting of signal transduction and DNA repair enzymes sensitizes PDAC cell cultures to photon and proton irradiation. (**A**) 3D tumoroid formation capacity of unirradiated Colo357 and MiaPaCa-2 cells 1-h pretreated with indicated pharmacological inhibitors (experimental set-up shown in Appendix A); (**B**) Normalized 3D PDAC tumoroid formation capacity upon 1-h inhibitor pretreatment combined with 6-Gy photon or proton irradiation. Data in the order of cell-line-specific inhibitor efficacy after photon irradiation. Enhancement ratio (tumoroid formation capacity after 6 Gy control treatment/tumoroid formation capacity after 6 Gy inhibitor treatment) are indicated as blue triangles; (**C**) Differences in the radiosensitizing efficacy of inhibitors visualized by Δ values of normalized tumor formation capacity (tumoroid formation capacity after 6 Gy of protons - tumoroid formation capacity after 6 Gy of photons). All results show mean ± SD (*n* ≥ 2); Δ: delta, i: inhibitor, TF: tumoroid formation.

**Figure 5 cancers-12-03216-f005:**
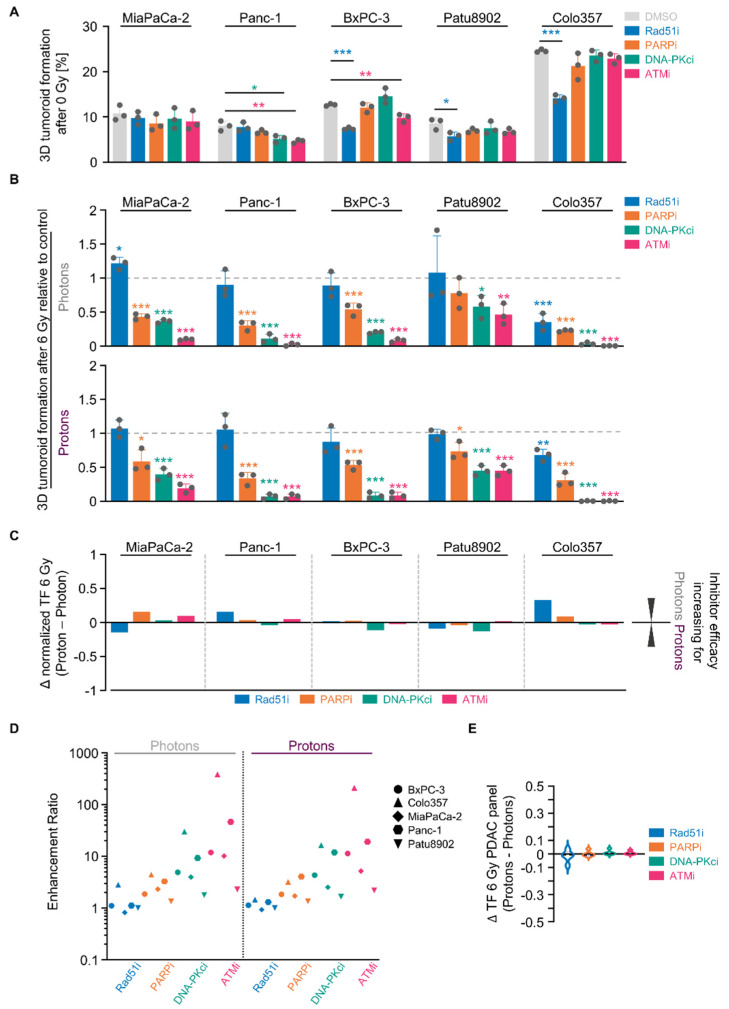
Targeting NHEJ-associated enzymes seems generally potent to sensitize 3D PDAC cell cultures to photon and proton irradiation. (**A**) 3D tumoroid formation capacity of unirradiated indicated PDAC cells treated with indicated DNA repair inhibitors (experimental set-up shown in Appendix A); (**B**) Normalized 3D PDAC tumoroid formation capacity upon 1-h inhibitor pretreatment combined with 6-Gy photon or proton irradiation; (**C**) Differences in in the radiosensitizing efficacy of inhibitors visualized by Δ values of normalized tumor formation capacity (tumoroid formation capacity after 6 Gy of protons - tumoroid formation capacity after 6 Gy of photons); (**D**) Enhancement ratio (tumoroid formation capacity after 6 Gy control treatment/tumoroid formation capacity after 6 Gy inhibitor treatment) of each cell line analyzed in B (see Appendix A) after photon and proton irradiation plotted logarithmically; (**E**) Analysis of differences in the radiosensitizing efficacy of indicated inhibitors showing violin blots of summarized Δ values of tumoroid formation capacity from all PDAC cell lines analyzed in Appendix A. All results show mean ± SD (*n* = 3; two-sided *t*-test; *, *p* < 0.05; **, *p* < 0.01; ***, *p* < 0.001); Δ: delta, TF: tumoroid formation.

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
