# Peer review of "Comparative Proton and Photon Irradiation Combined with Pharmacological Inhibitors in 3D Pancreatic Cancer Cultures"

_cancers, 2020, doi:10.3390/cancers12113216_

Round 1

Reviewer 1 Report

Dear Editor, 

I re-reviewed the manuscript by Goerte et al. I regret by the authors did not improve the clarity of their manuscript. They haven't adressed most the my concerns (see details below). For me, this manuscript cannot by published in this present form. 

Reviewer 1:

In this manuscript, the authors attempt to prove the better efficacy of proton therapy versus photon therapy in 3D pancreatic cancer models without additional efficacy when combined with DNA repair inhibitors. They further investigate the effect of both therapies on phosphoproteomes and demonstrate that proton therapy displays a greater effect on phospho-modifications.

Although the findings are potentially interesting for the field, the description of the results is extremely confusing and hard to follow. I think this manuscript requires a profound re-writing of the Results section to make this study understandable for Cancers readers. I regret but I have no other options than suggesting the rejection of this manuscript.

Results:

Major points:

Figure 1A:

I have a little concern regarding the 3D pancreatic cancer model. The authors mention in their title “3D pancreatic cancer tumoroids”. However, they treat with protons or photons the PDAC-derived cells only 24h after their seeding in lrECM. At this stage, I assume that the cells have not formed 3D tumoroids. Therefore, is it correct to utilize in 3D tumoroids in the title ? When I read for the first time the title, I was expecting that the authors treat with proton therapy the 3D tumoroids after they were formed. What they measure is the growth of 3D tumoroids (what they mention in the Results). I think the title should be slightly modified according to this comment.

Thank you very much. In the text we explain that in all our experiments we evaluate the tumoroid formation capacity/growth. In our opinion, after reading our experimental set-up or Fig 1A, readers easily understand that single cells were treated in 3D and tumoroid formation capacity was measured after a cell line-dependent growth period – as generally done for clonogenic assays. We changed the title using cultures instead of tumoroids and exchanged the term “tumoroid” with “cell culture” where appropriate throughout the manuscript.

Reviewer response: It is fine for me.

Figure 1C: the description of this figure and the figure itself is not clear and confusing for me. It took me a while to understand the graph presented in the figure were in absence of photon or proton therapy whereas it is written “photon and proton” in the legend. In addition, the description of the results (page 2, line 75 to 80) has to be clarified. For instance, the sentence “ the degree of radiosensitivity deviated from photon to proton irradiation” is not clear although I kind of understand what the authors meant.

Thank you. In Fig. 1C, we show that the different experimental set-ups of photon and proton irradiation do not impact basal tumoroid formation capacity. We changed the sentence in line 80/81 as follows: “In general, the degree of radiosensitivity towards photon irradiation differs to the one found for proton irradiation”.

Reviewer response: Your sentence in your response “we show that the different experimental set-ups of photon and proton irradiation do not impact basal tumoroid formation capacity” would be have been better to my opinion.

Figure 2: In this figure, the authors measure the effects of photon and proton therapy on the phosphoproteome of MiaPACA-2 and Colo357 cell lines. They observed more phospho-modifications with proton therapy than with photon therapy. It is sad that these experiment were performed only on 2D cell lines. It could have been interesting to obtain these results from 3D tumoroids. Moreover, the description of the results (page 3, line 102 to 105) is once again difficult to understand at the first reading. For instance, “ 6 out of 37 and 10 out of 34….”, we have no idea of what these 37 and 34 proteins are. The authors should better describe their results.

We are grateful for your comment. Both cell lines were always treated in 3D, also for phosphoproteome analysis as mentioned in the figure legend and in the Material and Method Section. Before irradiation and harvesting the cells, they were cultured in 3D.

Thank you for pointing out an unclear description in our results part. We changed “sites” to “protein phosphorylation sites” in line 106.

Reviewer response: I am sorry but the change “sites” to “protein phosphorylation sites” in line 106 does not make the description of your results any clearer. Moreover, there is no statistical analyses of these data. Out of 331 detected proteins and 584 phosphorylation sites, what would have been the probability to obtain similarities only by chance ?

Then, the authors attempt to identify phosphorylation changes (>50%) that are specific of proton and photon therapy in their two cell lines. Once again the description of the results is confusing. They identified few modification sites that are specific for both treatments but since the phosphorylation inhibits the enzymatic activity, induces ubiquitination or lacks specific inhibitors, they decided to analyze their data on the protein level instead of phosphorylation level. Where is the description of this analysis (Fig 2F) ?

We are grateful for indicating unclarities. The figure legends (Fig. 2F and S1) provide the information of the analyses of our data by the use of Venn diagrams. Protein names without phospho-sites were used.

Reviewer response: I regret but I do not see the findings obtained from the Figure 2F. What did the authors interpret from this analysis except that they have 3 proteins in common, for instance, between Colo357 and MiaPACA-2 after photon therapy. Once again, what would been this number only by chance (no statistical analysis) ?  

Figure 3: In this part, the authors selected several modified phosphorylation sites after photon and proton therapy. They treat cell lines with corresponding inhibitors to potentiate the effects of proton and photon therapies. There is a lack of precision in the description of Figure 3B. The authors claim that “they found only non-significant effects in both analyzed cell lines”. I am sorry but I see on the Figure 3B an asterisk (corresponding to a p value < 0.05) in tamoxifen treatment of Colo357 cell line.

Thank you for pointing out our mistake. We changed “non-significant” to “moderate” effects in line 143.

Reviewer response: It is fine to me.

The Figure 3C is extremely hard to understand. The y-axis of the left seems to correspond to the 3D tumoroids formation after 6 Gy relative to control. The y-axis on the right corresponds the enhancement ratio. To what correspond the triangles on the graph ? More importantly, the values described in the text (for instance, by 1.2- and 1.6-fold for tamoxifen in MiaPACA-2) does not correspond to anything I can see on the graph. This must be modified and clarified.

Thank you for your question. The triangles indicate the values of the enhancement ratios (as written in the Result Section). The relationship between triangles and second y-axis is also underscored by matching colors of triangles and right y-axis title.

Reviewer response: I am sorry but the fact that “triangles indicate the values of the enhancement ratios” is not mentioned in the Results section (neither in the Figure legend) or tell me which lines.

The sentence page 5 line 149 “Our findings suggest targeting of E-alpha and Chk1 to convey slightly higher radiosensitizing efficacy to photons than protons” makes no sense to me. Please clarify.

Thank you. By calculating the Δ values (tumoroid formation of 6-Gy protons (TF 6-Gy) minus TF 6-Gy photons; see figure legend), we show a difference between the drug-related radiosensitizing potential for photons and protons. In this case, we see Δ values ≥ 0 meaning that the drug mediates a greater sensitization to photons than protons.

Reviewer response: I had understood the meaning of this sentence but thank you for the explanation. My point was that your sentence makes grammatically no sense.

The authors could have validated some of their phosphorylated changes by Western blot analysis with corresponding phosphorylated forms-directed antibodies (if available).

Thank you. Lysates of both cell lines, Colo357 and MiaPaCa-2, were analyzed for expression levels of total and phosphorylated (p-) Chk1 (phospho-site S286), -ER-α (phospho-site S118) and -HER2 (phospho-site Y1248) after 0 Gy and after irradiation with 6-Gy photons or protons. Antibodies for activating phospho-sites were chosen, as the affected phospho-sites revealed by phosphoproteome analysis differ between the two cell lines. Basically, no changes in phosphorylation after irradiation with photons or protons were detected for p-ER-α (S118) and p-HER2 (Y1248), whereas p-Chk1 (S286) increases after both types of irradiation.

Reviewer response: The quality of Western blot for the validation are not of great quality. More importantly, the authors have the honesty to specify the blots they picked to assemble to final Figure. However, the cannot pick blots from different experiments to make the final one (the blot of protein of interest with the loading control of another experiment even though loaded on the same membrane).  

Figure 4B:

Once again, the description of the results is rather confusing. I do not see the meaning of the triangles on the graphs.

Thank you. We have addressed above (Fig 3).

Reviewer response: same response as above.

Minor points:

Page 2 line 75: the acronym lrECM is not explained when it first appears in the text. Please modify.

Thank you. We modified it accordingly and put the spelled out word “laminin-rich extracellular matrix (lrECM)” in the Results Section line 75/76.

Reviewer response: thank you

Page 2 line 79: same comment for SF2

Thank you. We modified it accordingly and put the spelled out “surviving fraction of 2 Gy (SF2)” word in the Results Section line 79.

Reviewer response: thank you

Page 7 line 176: MRNcomplexi ?

Thank you. We modified it accordingly and put the spelled out word “Mre11, Rad50 and Nbs1 complex (MRNcomplex)” in the Results Section line 181.

Reviewer response: thank you

Reviewer 2 Report

Review comments of cancers-978077

The authors have reported the manuscript entitled “Comparative proton and photon irradiation combined with pharmacological inhibitors in 3D pancreatic cancer cultures” and the experimental results were supported by their findings. I would recommend this manuscript for publication after full fill the following minor comment. In the revised manuscript, authors have not addressed the following comment.

  1. It would be better to have the fluorescence imaging to the live and dead cell staining assay of the 3D PADC tumoroid growth formation upon Photon and Proton irradiations.

Author Response

Comments to Reviewer 2:

The authors have reported the manuscript entitled “Comparative proton and photon irradiation combined with pharmacological inhibitors in 3D pancreatic cancer cultures” and the experimental results were supported by their findings. I would recommend this manuscript for publication after full fill the following minor comment. In the revised manuscript, authors have not addressed the following comment.

  1. It would be better to have the fluorescence imaging to the live and dead cell staining assay of the 3D PADC tumoroid growth formation upon Photon and Proton irradiations.

Thank you. From the scientific point of view, we think, as we described in Revision round 1, that the aspect of (early) cell kill induced by photons and protons is a huge topic requiring a particular set of methods and experiments. Further, cells from solid tumors, in contrast to leukemic cells, usually fail to undergo apoptosis or similar early cell death processes. We know that cells can recover from measurable events that seem to indicate that a cell is dead. Clonogenic survival and the consecutive formation of colonies provide a much more robust readout whether a cell could maintain its proliferative capacity (= tumoroid formation capacity/clonogenic survival) upon irradiation or not. In the context of your study, this readout is beyond the scope of our manuscript.

Of note, the experimental proton irradiation at our institution is currently not possible due to maintenance and other factors. The experimental beam cannot be used during the next three to four months.

Reviewer 3 Report

Please amend Line 399/400 to - "To determine differences in the early events of DNA damage between proton and photon irradiation, whole cell lysates were harvested 1 hour post-treatment as previously published [43]" - in line with the response to reviewer. 

Line 405. The cut off is arbitrary (author decided). The comment at line 407 isn't required. 

Typographical:

Line 344 "laminin-rich extracellular matrix (lrECM;(Matrigel™; BD, Heidelberg, Germany). lrECM already abbreviated at line 75-76.

In Figure 3,4,5 legends "tumoroids" has been replaced with "cell culture". But not in the remaining caption or elsewhere in the manuscript. The terminology should be consistent.

All other changes are adequate.

Author Response

Comments to Reviewer 3:

Please amend Line 399/400 to - "To determine differences in the early events of DNA damage between proton and photon irradiation, whole cell lysates were harvested 1 hour post-treatment as previously published [43]" - in line with the response to reviewer. 

Thank you. We changed line 400-402 as follows: "To determine differences in the early events on the molecular level, here the DNA damage response upon proton and photon irradiation, whole cell lysates were harvested 1 hour post-treatment as previously published [43]"

Line 405. The cut off is arbitrary (author decided). The comment at line 407 isn't required. 

Thank you. We changed line 408 as follows: "Proteins with phosphorylation site changes of at least 30% decrease or 50% increase (arbitrary cut-off) were considered relevant, selected and changes in both cell lines comparatively analyzed.”

Typographical:

Line 344 "laminin-rich extracellular matrix (lrECM;(Matrigel™; BD, Heidelberg, Germany). lrECM already abbreviated at line 75-76.

Thank you. We deleted the redundant abbreviation in line 345.

In Figure 3,4,5 legends "tumoroids" has been replaced with "cell culture". But not in the remaining caption or elsewhere in the manuscript. The terminology should be consistent.

Thank you. In line 307, in the lines 429, 430-432, 434 and in the Figure Legends of Supplement Figures S2, S3, S5 and S6 we replaced “tumoroids” with “cell cultures”. In the other parts of our manuscript, whenever tumoroid formation or growth is used, the term tumoroid is necessary and correct.

All other changes are adequate.

This manuscript is a resubmission of an earlier submission. The following is a list of the peer review reports and author responses from that submission.

Round 1

Reviewer 1 Report

In this manuscript, the authors attempt to prove the better efficacy of proton therapy versus photon therapy in 3D pancreatic cancer models without additional efficacy when combined with DNA repair inhibitors. They further investigate the effect of both therapies on phosphoproteomes and demonstrate that proton therapy displays a greater effect on phospho-modifications.   

Although the findings are potentially interesting for the field, the description of the results is extremely confusing and hard to follow. I think this manuscript requires a profound re-writing of the Results section to make this study understandable for Cancers readers. I regret but I have no other options than suggesting the rejection of this manuscript.   

Results:

Major points:

Figure 1A:

I have a little concern regarding the 3D pancreatic cancer model. The authors mention in their title “3D pancreatic cancer tumoroids”. However, they treat with protons or photons the PDAC-derived cells only 24h after their seeding in lrECM. At this stage, I assume that the cells have not formed 3D tumoroids. Therefore, is it correct to utilize in 3D tumoroids in the title ? When I read for the first time the title, I was expecting that the authors treat with proton therapy the 3D tumoroids after they were formed. What they measure is the growth of 3D tumoroids (what they mention in the Results). I think the title should be slightly modified according to this comment.

Figure 1C: the description of this figure and the figure itself is not clear and confusing for me. It took me a while to understand the graph presented in the figure were in absence of photon or proton therapy whereas it is written “photon and proton” in the legend. In addition, the description of the results (page 2, line 75 to 80) has to be clarified. For instance, the sentence “ the degree of radiosensitivity deviated from photon to proton irradiation” is not clear although I kind of understand what the authors meant.  

Figure 2: In this figure, the authors measure the effects of photon and proton therapy on the phosphoproteome of MiaPACA-2 and Colo357 cell lines. They observed more phospho-modifications with proton therapy than with photon therapy. It is sad that these experiment were performed only on 2D cell lines. It could have been interesting to obtain these results from 3D tumoroids. Moreover, the description of the results (page 3, line 102 to 105) is once again difficult to understand at the first reading. For instance, “ 6 out of 37 and 10 out of 34….”, we have no idea of what these 37 and 34 proteins are. The authors should better describe their results.

Then, the authors attempt to identify phosphorylation changes (>50%) that are specific of proton and photon therapy in their two cell lines. Once again the description of the results is confusing. They identified few modification sites that are specific for both treatments but since the phosphorylation inhibits the enzymatic activity, induces ubiquitination or lacks specific inhibitors, they decided to analyze their data on the protein level instead of phosphorylation level. Where is the description of this analysis (Fig 2F) ?

Figure 3: In this part, the authors selected several modified phosphorylation sites after photon and proton therapy. They treat cell lines with corresponding inhibitors to potentiate the effects of proton and photon therapies. There is a lack of precision in the description of Figure 3B. The authors claim that “they found only non-significant effects in both analyzed cell lines”. I am sorry but I see on the Figure 3B an asterisk (corresponding to a p value < 0.05) in tamoxifen treatment of Colo357 cell line.

The Figure 3C is extremely hard to understand. The y-axis of the left seems to correspond to the 3D tumoroids formation after 6 Gy relative to control. The y-axis on the right corresponds the enhancement ratio. To what correspond the triangles on the graph ? More importantly, the values described in the text (for instance, by 1.2- and 1.6-fold for tamoxifen in MiaPACA-2) does not correspond to anything I can see on the graph. This must be modified and clarified.

The sentence page 5 line 149 “Our findings suggest targeting of E-alpha and Chk1 to convey slightly higher radiosensitizing efficacy to photons than protons” makes no sense to me. Please clarify.

The authors could have validated some of their phosphorylated changes by Western blot analysis with corresponding phosphorylated forms-directed antibodies (if available).

Figure 4B:

Once again, the description of the results is rather confusing. I do not see the meaning of the triangles on the graphs.

Minor points:

Page 2 line 75: the acronym lrECM is not explained when it first appears in the text. Please modify.

Page 2 line 79: same comment for SF2

Page 7 line 176: MRNcomplexi ?

Reviewer 2 Report

Review comments of cancers-900311

The authors have reported the manuscript entitled “Comparative proton and photon irradiation combined with pharmacological inhibitors in 3D pancreatic cancer tumoroids” and the experimental results were supported by their findings. The manuscript has reached the cancers MDPI journal standard and I would recommend this manuscript for publication after full fill the following minor comments.

  1. Authors have to correct the name as “non-homologous end joining” of the abbreviation of NHEJ.

  1. It would be better to have fluorescence imaging to do live and dead cell assay of the 3D PADC tumoroid growth formation upon Photon and Proton irradiations.

Reviewer 3 Report

The authors presented their studies in comparing the biological effects of some pancreatic cancer cell lines from photons and passive scattering protons using 3D culture technique with pharmacological inhibitors. Numerous data are presented in this report and many information of the DNA repair inhibitors effects are provided. However, the authors ignored the role of proton beam characteristics in the biological effects. The proton experiments are conducted in the middle point of an SOBP from a passive-scattering beam line. The authors should mentioned that the linear energy transfer (you can used dose-averaged LET) at the mid-SOBP is as low as 2 or 3 keV/um. Many experiments have revealed that at the low-LET value, the difference in biological effects between protons and photons are small. But for high-LET (> 10 keV/um), such as at Bragg peak and the distal edge, that is a different story. The difference in biological effects can be enhanced greatly.

The data in the present study is sufficient to be published, but the authors must emphasized the conclusions are made based on the low-LET proton therapy at the mid-SOBP. And say something like the high-LET effects will be conducted in the future.

Some specific comments are listed as follows.

(1) abstract, there is a typo “endjoing”

(2) 4.1 cell culture, please spell out ATCC even though it is known by our community.

(3) 4.3.1, correct “200-kV” to ”200-kVp” (here p stands for the peak voltage the x-ray tube is 200 kV)

(4) 4.3.2, the proton beam field size is only 10 by 10 cm2. I don’t think it is large enough to deliver the uniform dose to the whole 96-well plate. A typical diameter of a well is about 9 mm, so the side of 12 wells is 10.8 cm. So, the data in column 1 and 12 may not be acceptable. I suggest the authors to increase the field size to cover the whole plate in your future experiments.

(5) 4.3.3, this statement is inappropriate. “the doses (D) were calculated with the polynomic formular:” The whole data processing is first fitting the data points in the linear quadratic form and from the fitting curve to get alpha and beta. Then you can solve D from the SF curve with known alpha, beta and the specified SF, i.e., 0.5 in your study.  Please rewrite this section and make it clear. By the way, formular is a typo, and it should be formula.

(6) 4.3.2, please ask your physics team to provide you the calculated dose-averaged LET at the middle point of the SOBP. If not available, you can find an estimated value from literature and introduce it here. The LET effect is very important in proton and heavy ion therapy.

(7) Please add below citation to your paper, which shows the LET effect in biological effects.

Guan, Fada, et al. "Spatial mapping of the biologic effectiveness of scanned particle beams: towards biologically optimized particle therapy." Scientific reports 5.1 (2015): 1-10.

(8) Please explain the difference between the tumoroids in your study and the clonogenics in traditional radiation biology experiments.

Reviewer 4 Report

Goerte and colleagues use 3D PDAC tumoroids to identify pharmacological inhibitors to radioenhance proton and/or photon irradiation. They identify that that combined irradiation and blockade of NHEJ DNA repair proteins may provide a therapeutic avenue. While the notion of combining DNA repair inhibitors with radiotherapy itself is not novel with a number of phase I/II clinical trials underway, the application to PDAC, 3D tumoroids, and comparison of proton vs photon therapy is novel and of current interest.

The manuscript is well-written and figures clearly presented. However, could benefit from 1-2 additional experiments.

Comments below:

Results:

Line 76. pH2AX expression in the tumoroids following irradiation using fluorescent microscopy would add to the proton vs photon cytotoxicity by validating more dsDNA strand breaks in PBT. And therein linking into the DNA repair inhibitor aspect. While this is often shown in 2D cultures, the authors are in a unique position to show it in 3D PDAC tumoroids with RT/PBT. 

Line 93. Due to the highly dynamic phosphorylation processes a temporal phosphoproteome profile would have been more insightful. A number of phos events that may have been dynamically changing are lost at this 1 hour post-irradiation snapshot and applied >50% increase / >30% decrease. Please add the following information into the method section to clarify:

  1. Why was 1-hour post-irradiation used?
  2. Why/how was the >50% increase / >30% decrease cut-off selected?

Line 145/Fig 3/Fig S2: Given the 1.7-fold change in pHER2 following proton irradiation but failure of Trastuzumab:

  1. Can the authors provide protein data to show the 'absolute expression' of total & p ERa, HER2 and Chk1 in the two cell lines as this varies greatly among PDACs?
  2. Can the authors repeat the HER2 experiment with a HER2 small molecule inhibitor (e.g. Lapatinib or Afatinib)? As a mAb, the in vitro absence of Trastuzumab cytotoxicity may be due to a lack immune (T) cells. Effector cells are required to induce antibody-dependent cell-mediated cytotoxicity of Trastuzumab in PDAC cells (https://clincancerres.aacrjournals.org/content/12/16/4925). The authors may wish to add this point to their discussion (line 263-265).

Fig 2C. Waterfall plot x-axis should be a log2 scale since a 2-fold increase (2) and 2-fold decrease (0.5) are equivalent changes. In other normalised graphs (e.g. 3C, 4B, 5B, etc) the y-axis also should be log2, for the above reason.

Fig 2F/Fig S1. These figures appear to be a duplicate, except the numbers in the Venn diagram differ slightly with no clear explanation why. Notably the legends are almost identical. How do these figures differ? - other than the naming of the shared up-regulated phospho-sites, which could be added to Fig 2F - and Fig S1 deleted.

Line 175. “PI3Ki showed the strongest effect in both cell lines”. This result is followed by the comment in the discussion “PI3K inhibition, still under debate as potential cancer target [26], did sensitize PDAC cells to photon and proton irradiation [27].” (Line 283-284) However, PI3Ki (LY294002) is also used as a broad autophagy inhibitor, with autophagy being a key survival mechanism of cancer cells following RT/PBT. Suggesting a future avenue of research for autophagy inhibitors in radioenhancement of RT/PBT.

Methods:

Line 322. “All cells were tested negative…”

Discussion:

Line 237-238/258-259. I think you need to qualify as “1 hour post-irradiation” as this is a very dynamic process and you have only a snapshot. Many protein are biphasic, if not multiphasic in response to drug/RT. This is perhaps the more important than the number of proteins on the array and limited cell lines screened.

Line 263-265. Modify in line with the above comment for Line 145/Fig 3/Fig S2.

Line 280-282. I disagree, this contrast is likely due to the use of a mAb vs small molecule inhibitor and the requirement for immune cell presence for the former (antibody-dependent cell-mediated cytotoxicity). This is not be true of all mAb, but appears to be true for Tras in PDAC cell lines.

Line 283-284. PI3Ki (LY294002) is also used as a broad autophagy inhibitor, with autophagy being a key survival mechanism of cancer cells following RT/PBT. Suggesting a future avenue of research for autophagy inhibitors in radioenhancement of RT/PBT.

Line 289-291. Again, this is limited to your arbitrary cut offs and a 1 hour post-irradiation time point.

Line 294. Word choice “…addiction…” – subjective term.

Conclusion:

Line 386. “…the latter leading to more critical changes at one hour post-irradiation”. I think you need to qualify this.
